# OpenReview forum: "LiveNewsBench: Evaluating Web Search Agents with Freshly Curated News"
_ICML.cc/2026/Conference — ICML 2026 regular_

### Official Review · Reviewer_BG75 · 2026-03-10

**Soundness:** 3
**Presentation:** 3
**Significance:** 3
**Originality:** 2
**Overall Recommendation:** 3
**Confidence:** 4

**Summary:**

This paper introduces LIVENEWSBENCH, an automatically updated benchmark for evaluating LLM agentic web search capabilities. It generates multi-hop Q&A pairs from recent news articles via an LLM-based pipeline, leveraging temporal freshness to separate internal knowledge from search ability. The authors evaluate 13 LLMs and 2 commercial search APIs, demonstrating wide performance variation (11%–90%) and conducting ablation studies on search budgets and memorization.

**Compliance With Llm Reviewing Policy:**

Affirmed.

**Final Justification:**

LIVENEWSBENCH addresses a well-motivated problem: disentangling LLM internal knowledge from agentic search capability, with a scalable automated pipeline and strong discriminative power. The rebuttal provided useful clarifications, including training cutoff dates (resolving W1), dataset statistics (W5), and an evaluation of o4-mini-deep-research (partially addressing W3).

However, two concerns remain only partially resolved. First, regarding W2, while I acknowledge the difficulty of precise training data attribution, the authors did not engage with the suggested approach of using existing query tools (e.g., infini-gram) to inspect the 13 OLMo correct answers — a tractable analysis that would meaningfully strengthen the anti-memorization claims. Second, regarding W3, the Deep Research evaluation was limited to a commercial report-oriented system (o4-mini-deep-research), which is not designed for concise factual QA. Recent open-source deep-research-style agents oriented toward short-form information seeking (e.g., WebSailor, WebShaper, WebDancer) would provide a fairer and more informative comparison. Additionally, the limited technical novelty (W4) and single-model pipeline dependency remain inherent limitations.

I maintain my score of 3 (Weak Reject). The paper has clear merits, but the incomplete analysis of no-internet accuracy and the narrow scope of evaluated search agents leave important gaps for a benchmark paper at this venue.

**Key Questions For Authors:**

1. The paper provides only a single real dataset example (Figure 1). For a benchmark paper, this is insufficient for readers to assess the quality, diversity, and difficulty distribution of the questions. Could you include a more representative set of examples in the appendix, covering different news domains, difficulty levels, and question types?

2. Please see the Weaknesses section.

**Limitations:**

Partially. The paper discusses future work on temporally evolving answers and acknowledges that models can still guess some answers without search (Section 5.5). However, several important limitations are not discussed: (1) the single-model dependency on GPT-5.1 for generation and verification, (2)  the difficulty of verifying anti-memorization claims for closed-source models whose training cutoffs are undisclosed — the authors should at least acknowledge this as a fundamental limitation of the evaluation, and (3) the English-only and news-only scope of the benchmark. The societal impact statement (Section 7) is adequate.

**Strengths And Weaknesses:**

Strengths

S1. Well-motivated problem. Disentangling internal knowledge from search capability is a genuine challenge. The paper demonstrates this effectively — Gemini 3 Pro achieves 74.3% on FreshQA without internet but only 20.5% on LIVENEWSBENCH (Table 2), clearly justifying the need for a new benchmark.

S2. Scalable automated pipeline. The end-to-end pipeline from Wikipedia Current Events through article retrieval, LLM-based Q&A generation, self-consistency verification, and guideline adherence checking is well-designed and cost-effective (~$700 per release). Table 4 shows accuracy differences between the human-verified and full test sets are within 5.1%, validating the pipeline's reliability.

S3. Strong discriminative power. The multi-hop, "Google-Proof" design requiring synthesis across multiple articles is well-conceived. The search budget ablation (Table 5) directly confirms the multi-step nature: increasing budget from 1 to 7 yields 4%–64.5% absolute gains. The overall 11%–90% accuracy range demonstrates strong discriminative power.

S4. Controlled and comprehensive evaluation. The standardized ReAct-based framework with fixed search/visit budgets enables fair cross-model comparison. The paper covers a broad range of systems and surfaces useful findings, such as tool-calling reliability issues (Kimi K2 Thinking's 44% format failure rate) and models' preference for search snippets over full-page content.


Weaknesses

W1. Temporal overlap between test data and model training data. The paper claims "limited memorization" because questions are derived from events after models' training cutoffs, yet it does not report any model's training data cutoff date. GPT-5.2 was released on December 11, 2025, while the test set covers November–December 2025 news events. Without knowing GPT-5.2's training cutoff, its leading 21.5% no-internet accuracy (Table 2) could reflect partial memorization rather than pure reasoning.

W2. Shallow analysis of no-internet correct answers. Models achieve 3.5%–21.5% accuracy without internet, but the analysis in Section 5.5 offers only a single anecdotal example (CIA strike) attributed to "world knowledge reasoning." This is insufficient for a paper whose core contribution is disentangling internal knowledge from search capability. The authors could conduct a more systematic analysis by evaluating models with fully open training data (e.g., OLMo-2 or OLMo-3), which would allow direct inspection of whether correct answers stem from training data overlap, world knowledge reasoning, or lucky hallucinations. A breakdown of no-internet correct answers by source — what proportion reflects genuine reasoning, what proportion is coincidental guessing, and what proportion may trace to training data — would be far more informative than a single anecdote and is essential for validating the benchmark's anti-memorization claims.

W3. Missing evaluation of Deep Research systems. The paper claims to evaluate "agentic web search capabilities" but excludes Deep Research systems, both commercial (e.g., OpenAI Deep Research, Gemini Deep Research) and open-source (e.g., Tongyi-DeepResearch), which represent the strongest search agents with substantially higher search budgets. Table 5 already shows that more searches significantly improve performance, so rankings under a 5-search limit may not reflect models' true search capability ceilings. Moreover, the paper criticizes Deep Research benchmarks for subjective evaluation in Related Work, yet does not test Deep Research models under its own objective evaluation framework — precisely the gap LIVENEWSBENCH could fill. Including at least a few major Deep Research systems would significantly strengthen the paper's completeness.


W4. Limited technical novelty and single-model verification bias. Automated news-based QA benchmark generation is now well-established, and this paper's pipeline follows the standard paradigm without substantial technical contribution. More critically, the entire pipeline depends on a single model family: GPT-5.1 Thinking for Q&A generation, correctness verification, and guideline adherence checking, with GPT-4.1 as the sole evaluation judge. This risks systematically embedding GPT-specific biases into the benchmark — questions that GPT-5.1 struggles to generate or verify are disproportionately filtered out, potentially skewing difficulty distribution in ways that favor or penalize certain model families.

W5. Lack of dataset statistics. The paper provides minimal characterization of the dataset beyond aggregate counts. There is no analysis of topic distribution (e.g., politics, sports, science, business), question difficulty distribution, answer type distribution, or geographic coverage of news events.

---

> ### Author Rebuttal · Authors · 2026-03-31
>
> Thank you for your thorough and constructive feedback on LiveNewsBench. We would like to address your concerns below:
>
> > W1. Temporal overlap between test data and model training data.
>
> We would like to clarify that training data cutoffs are publicly available through model cards / documentations for all closed-source models evaluated in the paper, and none of these cutoffs overlap with our test set, which covers November-December 2025. The cutoffs are as follows:
>
> | Model | Training Cutoff |
> |---|---|
> | GPT-5.2 | Aug 2025 |
> | Claude Sonnet 4.5 | Sept 2025 |
> | Gemini 3 Pro | Jan 2025 |
> | Grok 4 | Nov 2024 |
>
> Since LiveNewsBench test set covers only events from November-December 2025, the test set is derived entirely from events after all models' training cutoffs, ensuring minimal temporal overlap.
>
> > W2. Shallow analysis of no-internet correct answers. The authors could conduct a more systematic analysis by evaluating models with fully open training data (e.g., OLMo-2 or OLMo-3).
>
> We evaluated OLMo-3.1 32B Thinking, considered one of the strongest open-data LLMs [1], on our 200-question human-verified test set offline. OLMo-3.1 achieved 6.5% accuracy (13/200). Given this small number of correct answers, a systematic breakdown would not yield statistically significant conclusions. We also note that even for open-data LLMs, attributing a specific model response to particular training samples is inherently difficult: OLMo 3's training corpus contains 6 trillion tokens [1], and LLMs are known to generalize well beyond verbatim matches, making precise attribution challenging [2,3].
>
> > W3. Missing evaluation of Deep Research systems.
>
> We initially excluded Deep Research systems because they are designed to produce long-form, comprehensive research reports (typically several thousand words), rather than concise factual answers derived through multi-hop search and reasoning (lines 47-55; 150-162). Additionally, Deep Research models are substantially more expensive to evaluate, as they typically reference a large number of sources.
>
> For this rebuttal, we evaluated OpenAI o4-mini-deep-research. We were unable to evaluate o3-deep-research or Gemini Deep Research Agent due to prohibitive costs (\$500–\$1,000 per model). o4-mini-deep-research achieved only **57.0%**, well below GPT-5.2's Official Web Search API score of **90.0%**, and also below 9 of the models evaluated in our local agent framework with a search and page-visit budget of 5. This confirms our intuition that Deep Research models are optimized for long-form synthesis and may underperform on factual, multi-hop QA tasks like those in LiveNewsBench, despite their significantly higher search budgets and costs.
>
> > W4. Limited technical novelty and single-model verification bias.
>
> During the design phase, we experimented with multiple models for question generation, including GPT-5 series, Claude Sonnet 4.5, Kimi K2 Thinking, and DeepSeek V3.2. Through human evaluation, we found that the GPT-5 series produced the most challenging, multi-hop questions needed to evaluate state-of-the-art LLMs on agentic web search. Other models tended to generate simpler, more straightforward questions, which would inflate benchmark scores. We further note that all OpenAI GPT series models are ranked below DeepSeek V3.2, Claude Sonnet 4.5, and Grok 4 in our controlled local search agent environment, so it is unlikely that this setup introduces significant systematic advantage for OpenAI GPT models.
>
> > W5. Lack of dataset statistics.
>
> We provide the statistics below:
>
> **Topic Distribution** (annotated by Wikipedia Current Events):
>
> | Category | % |
> |---|---|
> | Armed conflicts and attacks | 24.5% |
> | Politics and elections | 20.0% |
> | Law and crime | 19.5% |
> | International relations | 10.5% |
> | Disasters and accidents | 9.0% |
> | Business and economy | 7.0% |
> | Arts and culture | 3.5% |
> | Sports | 3.5% |
> | Health and environment | 2.0% |
> | Science and technology | 0.5% |
>
> **Geographic Distribution** (assessed by GPT-4.1):
>
> | Region | % |
> |---|---|
> | Asia | 32.5% |
> | Europe | 22.5% |
> | North America | 17.5% |
> | Africa | 12.5% |
> | South America | 7.5% |
> | Multiple continents | 5.5% |
> | Oceania | 2.0% |
>
> All answers are concise, objective, and factual, typically between 1-5 words. We prefer to not provide a difficulty breakdown, as difficulty is inherently subjective and hard to define in a principled way.
>
> > Could you include a more representative set of examples in the appendix?
>
> We will include additional examples in the camera-ready version. Our full dataset is available at the website and GitHub repo linked in the paper.
>
> We hope these clarifications address your concerns, and we will incorporate all discussions into the camera-ready version. We would be grateful if you would consider revising your score in light of this response.
>
> [1]: Olmo 3
>
> [2]: How much do language models memorize
>
> [3]: Language Models May Verbatim Complete Text They Were Not Explicitly Trained On

---

> > ### Author Rebuttal · Reviewer_BG75 · 2026-03-31
> >
> > Thank you for the detailed rebuttal. I appreciate the additional experiments and clarifications. I have two follow-up suggestions that may further strengthen the paper:
> >
> > Regarding W2 (Open-data model analysis):
> >
> > I understand that precise training data attribution is inherently challenging. However, I would like to point out that the OLMo team has developed query interfaces for searching their training corpus (e.g., infini-gram [1]). This tool allows n-gram and document-level searches over OLMo's pre-training data, which could help the authors perform a more targeted analysis of the 13 correct answers.
> >
> > Regarding W3 (Deep Research systems):
> >
> > I appreciate the evaluation of o4-mini-deep-research and the cost considerations noted. However, commercial Deep Research products (e.g., o3-deep-research, o4-mini-deep-research) are primarily designed for generating lengthy research reports, which may explain their underperformance on concise factual QA. Recent work from the open-source community has explored deep-research-style agents that are more oriented toward short-form question answering and information seeking, such as WebSailor [2], WebShaper [3], and WebDancer [4]. Evaluating such systems would provide a fairer comparison and a more complete picture of agentic search capabilities on LiveNewsBench.
> >
> > [1] https://huggingface.co/spaces/liujch1998/infini-gram
> >
> > [2] WebSailor: Navigating Super-Human Reasoning for Web Agent
> >
> > [3] WebShaper: Agentically Data Synthesizing via Information-Seeking Formalization
> >
> > [4] WebDancer: Towards Autonomous Information Seeking Agency

---

> > > ### Author Response · Authors · 2026-04-08
> > >
> > > We thank the reviewer for their thorough review and detailed feedback. We would like to address the questions below:
> > >
> > > > Regarding W2 (Open-data model analysis):
> > >
> > > Thank you for your suggestion. Following your feedback, we evaluated OLMo 2 0325 32B Instruct offline and attempted to attribute its correct answers to pretraining data using the Infinigram API. OLMo 2 32B answered only 8 out of 200 questions correctly. None of these 8 correctly answered questions could be matched to any documents, confirming the difficulty of using near-exact-match methods like Infinigram to attribute model behavior to pretraining data. We did note, however, that all 8 correct offline answers involved time intervals (e.g., "one week," "three hours") or place names (e.g., Jerusalem) that returned 100-10M+ matches in the pretraining dataset, suggesting the model may have simply guessed these answers by chance, given how frequently such tokens appear in training data.
> > >
> > > Unfortunately, the pretraining data for OLMo 3.1 32B Thinking (13/200 offline accuracy) is not available through the Infinigram API. Since this model's pretraining data amounts to 6T tokens, self-hosting Infinigram is not feasible for us, so we were unable to attribute its correct offline answers to pretraining data at this time.
> > >
> > > > Regarding W3 (Deep Research systems): Recent work from the open-source community has explored deep-research-style agents that are more oriented toward short-form question answering.
> > >
> > > Thank you for your suggestion on related open Deep Research models. We will incorporate a discussion of these models in the final version of the paper.
> > >
> > > Following your recommendation, we evaluated Tongyi Deep Research, an open-source deep research model, within our local ReAct agent implementation to assess whether such systems could be applied to search-based factual QA tasks like LiveNewsBench. The search budget was set to 15 search queries and 15 full-page visits, 3× the LiveNewsBench default, to accommodate the needs of Deep Research systems. We found that:
> > > 1. Tongyi Deep Research was unable to adapt to the prompts and tool specifications used in our local agent implementation, and therefore failed to complete approximately 50% of tasks successfully.
> > > 2. As a result, it correctly answered only 34.5% of questions, ranking 9th on our local ReAct agent leaderboard, between Kimi K2 Thinking and GPT-OSS 120B.
> > >
> > > We were unable to evaluate Tongyi Deep Research using its official harness for two reasons: (1) the official Tongyi Deep Research API is unavailable in our region due to regional restrictions, and (2) the full Tongyi Agent Harness requires six different search APIs as well as a code execution sandbox, making deployment costly and challenging for the rebuttal period.
> > >
> > > Finally, we noticed that your final justification was submitted prior to this rebuttal response. We would be grateful if you could revisit your final justification and overall rating in light of our responses to your remaining concerns. We deeply appreciate your thorough engagement throughout the review process.

---

### Official Review · Reviewer_gjac · 2026-03-12

**Soundness:** 3
**Presentation:** 2
**Significance:** 2
**Originality:** 2
**Overall Recommendation:** 4
**Confidence:** 3

**Summary:**

LiveNewsBench is a fully automated, quarterly-updated agentic web search benchmark specifically designed to address the limitations of existing QA benchmarks, namely their static and outdated nature as well as susceptibility to training data contamination.

**Compliance With Llm Reviewing Policy:**

Affirmed.

**Final Justification:**

The rebuttal is good. So I will keep 4.

**Key Questions For Authors:**

Please refer to weaknesses. Generally speaking, this work is quite good to be accepted.

**Limitations:**

yes

**Strengths And Weaknesses:**

Strength:
1.The automated, periodically updatable benchmark construction pipeline represents a significant contribution.
2.The time-splitting mechanism for preventing data contamination is an intriguing methodological design.
3.The human-verified subset effectively addresses concerns regarding reliability and annotation quality.

Weakness:
1.The search results for news can be influenced by biases in the search API being used. For example, using Jina's API versus Google Search will often produce different results.

2.There are already many similar works that use news articles to create datasets.

---

> ### Author Rebuttal · Authors · 2026-03-31
>
> Thank you for your thorough and constructive feedback on LiveNewsBench. We would like to address your concerns below:
>
> > The search results for news can be influenced by biases in the search API being used. For example, using Jina's API versus Google Search will often produce different results.
>
> We thank the reviewer for raising this important point. We fully acknowledge that different search APIs may introduce varying biases into retrieved results. However, within any given release of LiveNewsBench, all models are evaluated using the **same search engine**, ensuring that cross-model performance comparisons remain fair and consistent. Furthermore, LiveNewsBench's design naturally supports a secondary use case: by holding the LLM fixed and varying the search engine, practitioners can directly compare the relative performance of different search APIs, a capability we consider a feature rather than a limitation.
>
> > There are already many similar works that use news articles to create datasets.
>
> We appreciate the reviewer's feedback. While prior works have indeed leveraged news articles for dataset construction, LiveNewsBench constructs **complex, multi-hop queries** grounded in events that fall beyond model's pretraining cutoff, reducing the likelihood that models can answer via memorization alone. The benchmark is **regularly updated**, ensuring it remains relevant as state-of-the-art LLMs continue to advance. Together, these properties enable LiveNewsBench to cleanly evaluate *agentic search capability* as a distinct skill - a gap we believe existing news-based datasets do not adequately address.
>
> We hope these clarifications address your concerns, and we will incorporate all discussions into the camera-ready version. We would be grateful if you would consider revising your score in light of this response.

---

> > ### Author Rebuttal · Reviewer_gjac · 2026-04-01
> >
> > I have no further questions and will maintain my positive score.

---

> > > ### Author Response · Authors · 2026-04-08
> > >
> > > We appreciate your support for our work. Thank you again for your constructive and thoughtful feedback on our submission.

---

### Official Review · Reviewer_XZNt · 2026-03-12

**Soundness:** 3
**Presentation:** 3
**Significance:** 2
**Originality:** 3
**Overall Recommendation:** 4
**Confidence:** 3

**Summary:**

This work focuses on evaluating LLMs' agentic web search capabilities. Existing benchmarks often fall short because they either evaluate domain-specific reasoning rather than search (e.g., HLE), focus on long-form report generation (e.g., DeepResearch), or suffer from data contamination due to static questions that fall within the LLMs' training knowledge cut-offs. This paper proposes LiveNewsBench, a novel, regularly updated benchmark designed to genuinely test LLMs' agentic search abilities. By utilizing an automated pipeline to generate multi-hop Q&A pairs from fresh news articles, the benchmark ensures the questions require actual online retrieval rather than relying on memorized internal knowledge. Extensive experiments across various proprietary and open-source LLMs demonstrate the benchmark's discriminative power and provide insightful analyses regarding tool-calling budgets and the gap between offline and online performance.

**Compliance With Llm Reviewing Policy:**

Affirmed.

**Final Justification:**

I appreciate this paper's contribution in **constructing a highly valuable and challenging testbed for agentic search**, which remains difficult even for SOTA LLMs as of April 2026. The overall data construction methodology is intuitive, and the empirical results are comprehensive.

During the rebuttal phase, the authors' detailed response fully addressed my concerns regarding data construction costs, the performance ceiling of SOTA LLMs, and the impact of an extended tool-calling budget. Regarding further RL investigations, I completely understand the tremendous computational costs and the limited time window of the rebuttal. The authors' current efforts in dataset construction and evaluation already constitute a substantial contribution. This testbed will inspire future research into developing RL methods tailored to enhancing agentic search capabilities. Consequently, I am raising my score from a Weak Reject to a **Weak Accept** to **champion** this paper. I sincerely appreciate the authors' hard work during the rebuttal phase.

One minor suggestion: during the rebuttal, the authors observed that ***extending the tool-calling budget did not yield corresponding performance improvements***, as models tend to overconfidently output answers after only shallow rounds of search. I highly recommend incorporating this insightful analysis into the final manuscript. If applicable, I also suggest extending this discussion to include the most recent and capable deep-search agents, such as the ***MiroMind-1.7*** series, to compare the behavioral differences between specifically tuned search models and generic LLMs.

**Key Questions For Authors:**

Please refer to Weaknesses

**Limitations:**

Yes

**Strengths And Weaknesses:**

### **Strengths**

* ***S1 Timely research direction:*** Building a time-sensitive, contamination-limited benchmark for agentic search is highly critical. While multi-hop QA benchmarks are often already memorized by LLMs, and deep research benchmarks entangle search with long-form writing, this work provides a clean, objective, and verifiable testbed specifically for information-seeking capabilities.

* ***S2 Intuitive data construction:*** The automated data curation pipeline (from Wikipedia event extraction to LLM-based Q&A generation and self-consistency filtering) is logical, systematically designed, and well-explained. The inclusion of a human-verified subset further guarantees the reliability of the evaluation.

* ***S3 Comprehensive analysis:*** The experiments are extensive, covering a wide range of both proprietary and open-weight LLMs. The multifaceted analysis, including the impact of varying tool-calling budgets, comparisons between offline and online settings, and evaluations against other time-sensitive benchmarks, yields intuitive and valuable observations regarding LLM agentic behaviors.
---

### **Weaknesses**

* ***W1 Elaboration on curation and maintenance costs:*** While the authors briefly mention that the total cost of constructing the current release is $700 (Line 246), a more detailed breakdown would be highly beneficial. How much of this is attributed to API calls versus human verification? Given the commitment to quarterly updates, a discussion on the long-term sustainability, scalability, and computational costs of maintaining this "live" benchmark would strengthen the paper.

* ***W2 Upper bound of performance and tool-calling budget:*** The necessity of a strict tool-calling budget (5 searches, 5 visits) is understandable for standardizing evaluation, but it raises a question about the dataset's absolute difficulty. DeepSeek-V3.2-Thinking already reaches 84.5% accuracy under this constraint. Is the dataset challenging enough for the next generation of models? I am curious if providing an unlimited (or significantly larger) budget would allow top models to reach near 100% accuracy, or if there are inherent reasoning bottlenecks. Have the authors tested more recent models (e.g., Qwen3.5-series, Kimi-K2.5, and Seed-1.8/2.0) to see if the benchmark is already close to being saturated?

* ***W3 Practicality of task formulation and potential for RLVR:*** While I find the benchmark highly meaningful, I wonder about the real-world use case of executing such complex, multi-hop search trajectories merely to retrieve a short factual answer, compared to the more prevalent deep-research report generation. However, I see a massive potential here: as the authors briefly mentioned in the introduction, this scalable pipeline could generate data for RLVR. If the authors could provide a preliminary experiment, or even just a deeper discussion, showing that training an open-source model on this synthesized data significantly boosts its persistent information-seeking capabilities (and potentially transfers to deep research tasks), the impact of this work would be substantially amplified. For example, it would be highly compelling to discuss or demonstrate whether, under the same data quantity, models trained on LiveNewsBench's synthesized data could outperform those trained on existing synthetic datasets like ASearcher (which relies on iteratively rewriting multi-hop QA via entity obfuscation). Highlighting how this benchmark's pipeline can benefit data synthesis to actively boost agentic search capabilities would make the paper's contribution even stronger.

---

> ### Author Rebuttal · Authors · 2026-03-31
>
> Thank you for your thorough and constructive feedback on LiveNewsBench. We would like to address your concerns below:
>
> > W1 Elaboration on curation and maintenance costs.
>
> We would like to clarify that all of the 700 USD data curation cost reported in the paper stems from LLM API calls: approximately 100 USD was spent on verifying article relevance to news events, 200 USD on drafting candidate Q&A pairs, and 500 USD on subsequent correctness and guideline adherence verification steps. Human verification is performed by students conducting for-credit research at our institution, totaling 5-10 hours per release. Under our institutional policy, students engaged in for-credit research cannot receive monetary compensation, so no cost is attributed to human verification.
>
> > W2 Upper bound of performance and tool-calling budget
>
> > Is the dataset challenging enough for the next generation of models?
>
> Thank you for this question. To address it, we evaluated three leading models released after the ICML submission cycle: GPT-5.4 Thinking, Qwen3.5-397B Thinking, and Kimi K2.5 Thinking. We were unable to include Seed 2.0 due to regional restrictions. All evaluations used our local agent under the default budget (5 searches, 5 full-page visits).
>
> | Model | Score | Δ vs. Prior Version |
> |---|---|---|
> | GPT-5.4 Thinking | 62.0% | −12.0% vs. GPT-5.2 |
> | Qwen3.5-397B Thinking | 74.0% | +14.0% vs. Qwen3-235B Thinking |
> | Kimi K2.5 Thinking | 70.5% | +22.5% vs. Kimi K2 Thinking |
> | DeepSeek V3.2 Thinking *(prior SotA)* | 84.5% | N/A |
>
> All three recent models fall short of the prior state-of-the-art (DeepSeek V3.2 Thinking), and the mixed trend across models suggests that general capability improvements do not reliably translate to gains on LiveNewsBench, indicating the benchmark remains genuinely challenging.
>
> > Would a significantly larger search budget allow top models to approach 100% accuracy?
>
> Thank you for raising this important concern. To investigate, we re-evaluated the two top-performing models on our local agent, DeepSeek V3.2 Thinking and Claude Sonnet 4.5, under an extended budget of 15 searches and 15 full-page visits (3x the default).
>
> | Model | Default Budget Score | Extended Budget Score | Avg. Searches | Avg. Page Visits |
> |---|---|---|---|---|
> | DeepSeek V3.2 Thinking | 84.5% | 84.0% | 4.5 ± 2.9 | 2.8 ± 2.0 |
> | Claude Sonnet 4.5 | 82.0% | 81.0% | 4.0 ± 2.4 | 1.7 ± 1.5 |
>
> Despite the substantially larger budget, performance changed by less than 1% for both models, and neither approached 90-100% accuracy. This suggests that the primary bottleneck lies in query planning and reasoning, and simply increasing search budgets cannot overcome these limitations.
>
> > W3 Practicality of task formulation and potential for RLVR.
>
> Thank you for raising this point. We considered conducting a preliminary RLVR training study using the LiveNewsBench training set prior to submission. Unfortunately, the associated costs placed this experiment well beyond our budget.
>
> Even under conservative assumptions: 50 RL steps, 256 samples per step, 16 rollouts per sample, an average of 3 searches and 3 full-page visits per rollout, and a 32K context window, a single training run would require approximately 200K rollouts, 1.2M search API calls, and 6.4B tokens. At standard search API pricing (0.008 USD per call, e.g., Tavily), search costs alone would reach 10,000 USD. Training a 7-8B parameter model would further require ~750 H200 GPU hours (2,000 USD), bringing the total cost per run to approximately **12,000 USD**.
>
> Furthermore, because LiveNewsBench requires retrieval over diverse, recently published news articles, cost reduction via hosting a local web corpora (e.g., Common Crawl) is not straightforward: such dumps are frequently out of date and often lack dynamically rendered JavaScript pages, which constitute the majority of modern news sites.
>
> For these reasons, we focused our contribution on the data acquisition pipeline, automated Q&A construction and verification, and standardized evaluation metrics, and we leave RLVR training to the broader community and resource-rich research groups. We hope our dataset and pipeline serve as a useful foundation for such future work.
>
> We hope these clarifications address your concerns, and we will incorporate all discussions into the camera-ready version. We would be grateful if you would consider revising your score in light of this response.

---

> > ### Author Rebuttal · Reviewer_XZNt · 2026-04-03
> >
> > Thank you to the authors for addressing my concerns and conducting such extensive experiments. I have just one question: why does extending the tool-calling budget not yield proportional performance gains? Can this be attributed to the model's behavior, such as a tendency to overconfidently provide an answer after only shallow rounds of search? If so, would models specifically trained for Deep (Re)Search tasks, like the Tongyi-DeepResearch or MiroMind series, perform more persistent searches?
> >
> > Furthermore, based on your extensive experiments, I recognize the difficulty of running RL experiments. **I appreciate this effort and agree that LiveNewsBench serves as a valuable and challenging testbed for the community.**

---

> > > ### Author Response · Authors · 2026-04-08
> > >
> > > Thank you for your thorough and constructive feedback on LiveNewsBench. We address your questions below.
> > >
> > > > Why does extending the tool-calling budget not yield proportional performance gains? Can this be attributed to a tendency to overconfidently provide an answer after only shallow rounds of search?
> > >
> > > Thank you for raising this valuable question. We find that even with an extended budget (15 search queries and 15 full-page visits; 3x the default), leading models such as DeepSeek V3.2 Thinking and Claude Sonnet 4.5 do tend to provide answers overconfidently after only shallow rounds of search. As shown below, the average number of searches and page visits was well below the allocated budget, and very few questions triggered more than 10 tool calls of either type.
> > >
> > > | Model | Default Score | Extended Score | Avg. Searches | Avg. Page Visits | >=10 Searches | >=10 Page Visits |
> > > |---|---|---|---|---|---|---|
> > > | DeepSeek V3.2 Thinking | 84.5% | 84.0% | 4.5 ± 2.9 | 2.8 ± 2.0 | 7.5% | 0.5% |
> > > | Claude Sonnet 4.5 | 82.0% | 81.0% | 4.0 ± 2.4 | 1.7 ± 1.5 | 6.0% | 0.0% |
> > >
> > > We further analyzed correct vs. incorrect responses and found that tool usage is similar regardless of outcome. On average, Claude Sonnet 4.5 uses 3.9 searches and 1.5 page visits on correctly answered questions, compared to 4.5 searches and 2.6 page visits on incorrect ones. DeepSeek V3.2 Thinking similarly uses an average of 4.2 searches and 2.7 page visits on correct questions, versus 6.1 searches and 3.7 page visits on incorrect ones. The modest gap between these conditions suggests that today's models are indeed somewhat overconfident and do not leverage the additional budget to meaningfully improve their answer accuracy.
> > >
> > > > Would models specifically trained for Deep (Re)Search tasks, such as Tongyi DeepResearch or the MiroMind series, perform more persistent searches?
> > >
> > > For this rebuttal, we evaluated Tongyi-DeepResearch on our benchmark under the extended 15-search / 15-page-visit budget. It correctly answered 34.5% of questions, ranking 9th on our local ReAct agent leaderboard, between Kimi K2 Thinking and GPT-OSS 120B. Notably, it averaged only 4.03 searches and 1.74 page visits per question, a utilization rate comparable to non-deep-research models. This suggests that models specifically trained for deep research do not perform meaningfully more persistent searches on our factual QA tasks designed for agentic search.
> > >
> > > We would be grateful if you could revisit your review as well as overall rating in light of these responses. We sincerely appreciate your thorough and constructive feedback throughout the review process.

---

### Official Review · Reviewer_o2JE · 2026-03-12

**Soundness:** 3
**Presentation:** 3
**Significance:** 3
**Originality:** 2
**Overall Recommendation:** 3
**Confidence:** 4

**Summary:**

This study presents a novel, live benchmark called LiveNewsBench along with a scalable pipeline to auto-generate news articles and QA pairs to test web search abilities of state of the art models. The authors show high agreement of auto-generated tasks with human verified samples and further study model performance on this dataset.

**Compliance With Llm Reviewing Policy:**

Affirmed.

**Key Questions For Authors:**

I have no further questions than the ones listed in the weaknesses above. Addressing those directly impacts my score assigned

**Limitations:**

Yes

**Strengths And Weaknesses:**

Strengths:
- The evaluation of open ended web search tasks is hard due to overtraining of foundational models on the internet. A live benchmark for studying web search abilities is crucial need of the moment
- The generation pipeline is sound and high correlation with human agreement is a positive signal about the dataset

Weaknesses:
- Motivation behind selection of GPT-5.1 for data curation is unclear
- The authors consider this task as one that is distinct from deepresearch but the reason behind that is not clear to me. The authors show that larger number of web searches lead to better performance which contradicts their take on deepresearch.
- No human baselines are provided for LiveNewsBench. The authors should test human performance to verify that these tasks are really solvable under the provided constraints like limited web searches
- The benchmark appears to be saturated already: GPT-5.2 with its web search tool achieves a 90% on the benchmark leaving very little headroom for improvement
- The authors further prompt the model to specifically use less web searches during evaluation which biases the eval. If the authors want to define search budgets, this should be controllable from the environment side (deterministically) and not via prompting models to use lesser tool calls.

---

> ### Author Rebuttal · Authors · 2026-03-31
>
> Thank you for your thorough and constructive feedback on LiveNewsBench. We would like to address your concerns below:
>
> > Motivation behind selection of GPT-5.1
>
> During the experiment design phase, we evaluated multiple closed- and open-source models, including the GPT-5 series, Claude Sonnet 4.5, Kimi K2 Thinking, and DeepSeek V3.2. Human assessment determined that the GPT-5 series was most effective at generating challenging, multi-hop questions suited to evaluating state-of-the-art LLM agentic web search capabilities; other models tended to produce simpler, more straightforward questions that inflate benchmark performance. Since we also intend to evaluate the latest GPT-5 series model (GPT-5.2) and while avoiding self-evaluation bias, we chose to use GPT-5.1 to construct and validate our benchmark questions.
>
> > The authors consider this task as one that is distinct from deepresearch but the reason behind that is not clear to me
>
> LiveNewsBench targets factual questions with concise, verifiable answers, typically short phrases of 1-5 words. Deep research systems address a different task: they produce long-form reports of several thousand words that blend diverse facts with subjective analysis. Such reports are often excessive for user queries that simply require a direct answer. Moreover, long-form reports lack clear factual ground truth, making objective evaluation and RLVR training difficult. Our benchmark avoids this limitation. (lines 47–55; 150–162)
>
> Both deep research and factual QA benchmarks like LiveNewsBench benefit from multi-hop search and page visits, so it is unsurprising that increasing the search budget improves performance in both cases.
>
> > No human baselines are provided for LiveNewsBench.
>
> We conducted a preliminary human baseline study for this rebuttal. A human NLP researcher (not an author) attempted 50 questions from the human-verified test set under identical constraints: up to 5 web searches and 5 full-page visits, using Google Search with all LLM-assisted features disabled.
>
> | Metric | Human | Best LLM (DeepSeek V3.2 Thinking) |
> |---|---|---|
> | Accuracy | 50.0% | 84.5% |
> | Avg. searches used | 2.4 | 3.3 |
> | Avg. page visits used | 3.4 | 2.6 |
>
> A 50% human accuracy falls between Qwen3 235B A22B Thinking (9th, 60.0%) and Qwen3 8B (10th, 49.5%), well below the SotA LLMs. These results are unsurprising for two reasons. First, LLMs routinely exploit advanced search syntax, such as `site:` or quoted phrases to enforce must-include keywords, yielding more precise queries. Second, SotA LLMs are explicitly trained for agentic search via RLVR, and models optimized in this way are known to often match or exceed average human performance on well-defined, verifiable tasks. It is therefore expected that LLMs outperform a typical human researcher on this benchmark.
>
> > The benchmark appears to be saturated already.
>
> While GPT-5.2 achieves 90% accuracy using its official web search tool, we note that (1) this configuration uses an unconstrained search budget that is not configurable via API, and (2) the GPT-5.2 was likely specifically trained on the official web search tool, shrinking the train-test gap. Under our local agent with a limited search budget and a custom harness, the same model scores only 72%, suggesting that models struggle to generalize across different agentic search implementations and search budgets.
>
> Moreover, three leading models released after the ICML submission cycle show that the benchmark remains non-trivial: GPT-5.4 Thinking scores 62.0% (-12.0% vs. GPT-5.2), Qwen3.5-397B Thinking scores 74.0% (+14.0% vs. Qwen3-235B Thinking), and Kimi K2.5 Thinking scores 70.5% (+22.5% vs. Kimi K2 Thinking). These results suggest that general model improvements do not uniformly translate to gains on this benchmark. Further, all three models still fail short of previous state of the art results on the local agent (DeepSeek V3.2 Thinking, 84.5%), suggesting the benchmark is challenging even for newly released leading models.
>
> > The authors further prompt the model to specifically use less web searches … If the authors want to define search budgets, this should be controllable from the environment side (deterministically)
>
> We confirm that search and full-page visit limits are enforced **programmatically and deterministically** in our local agent implementation. We inform the model of the budget limits via prompting so it can plan its tool calls accordingly. We also include "use less web searches" in our prompt because not all queries require the maximum budget, and search efficiency is itself an evaluated dimension of our benchmark.
>
> We hope these clarifications address your concerns, and we will incorporate all discussions into the camera-ready version. We would be grateful if you would consider revising your score in light of this response.

---

> > ### Author Rebuttal · Reviewer_o2JE · 2026-03-31
> >
> > 1. I am not satisfied with the argument: "Human assessment determined that the GPT-5 series was most effective at generating challenging,.. other models tended to produce simpler, more straightforward questions that inflate benchmark performance". This claim needs statistical significance studies and well defined evaluation metrics
> >
> > 2. I disagree with the statement: "deep research systems address a different task: they produce long-form reports of several thousand words that blend diverse facts with subjective analysis." Deepresearch as a capability relies on a model doing exhaustive research and exploration which is independent of the output format and length. Several benchmarks such as DeepresearchQA[1], etc test these skills with constrained outputs while still prioritizing the exploration aspect.
> >
> > 3. The human baselines are not significant (agreement score calculation is impossible with one annotator). Furthermore, the poor human baseline here also indicates that the dataset quality might be poor in itself which is not addressed through the rebuttal.
> >
> > 4. In an ideal deepresearch task evaluation setting, restricting tool outputs is not the correct and most ecologically valid way to evaluate models. The authors' claim that the benchmark is saturated is not convincing to me because they condition this argument on restricting tool use. In the case where the human baseline reported also has this restriction, it provides clear signal about it being a poor setup.
> >
> >
> > [1] Gupta, Nikita, et al. "DeepSearchQA: Bridging the Comprehensiveness Gap for Deep Research Agents." arXiv preprint arXiv:2601.20975 (2026).

---

### Decision · Program_Chairs · 2026-04-30

**Decision:**

Accept (regular)

**Comment:**

This submission tackles an important and timely issue: distinguishing between what large language models know internally and their ability to perform genuine web searches as active agents. Existing static benchmarks struggle to do this effectively because of pretraining contamination. The work introduces an automated, quarterly-updated pipeline that generates multi-hop factual question-answer pairs from recent news events, complemented by a human-verified subset. It features controlled search budgets and tests a diverse range of systems, including commercial APIs, open-weight models, and a local ReAct framework.
The reviews agree that this is a solid, technically well-founded contribution. It offers a well-designed, cost-effective curation process, a reliable time-splitting method to avoid contamination, and strong discriminative performance, with accuracy ranging from 11% to 90%. The paper also provides useful insights into search budgets and tool usage.
In response, the authors provided a thorough rebuttal. They added documentation for closed-source models’ training cutoffs, conducted offline evaluations with OLMo and Infinigram attribution, tested both commercial and open-source research systems, performed an extended-budget experiment showing top models plateau well below perfect accuracy, included a human baseline, shared dataset statistics, and evaluated three leading models published after submission—all of which did not surpass previous state-of-the-art results. These findings directly address concerns about saturation.
In my view, the final scores do not fully reflect the strength of the rebuttal. The additional experiments convincingly addressed key technical points, and it seems one review was not updated after the authors’ responses. Judging by the technical substance rather than scores, this paper meets the standards for acceptance.
Contributions like this—automated, maintainable, contamination-resistant benchmarks designed to separate purposeful search from memorization—are especially valuable to the community, even if their methods are not groundbreaking. This work is carried out thoroughly and with care. I recommend accepting it.